# Discovery of a strain-stabilised smectic electronic order in LiFeAs

Chi Ming Yim [1], Christopher Trainer[1], Ramakrishna Aluru[1], Shun Chi[2,3], Walter N. Hardy[2,3], Ruixing Liang[2,3], Doug Bonn[2,3] & Peter Wahl [1]

In many high temperature superconductors, small orthorhombic distortions of the lattice structure result in surprisingly large symmetry breaking of the electronic states and macroscopic properties, an effect often referred to as nematicity. To directly study the impact of symmetry-breaking lattice distortions on the electronic states, using low-temperature scanning tunnelling microscopy we image at the atomic scale the influence of strain-tuned lattice distortions on the correlated electronic states in the iron-based superconductor LiFeAs, a material which in its ground state is tetragonal with four-fold ($C_4$) symmetry. Our experiments uncover a new strain-stabilised modulated phase which exhibits a smectic order in LiFeAs, an electronic state which not only breaks rotational symmetry but also reduces translational symmetry. We follow the evolution of the superconducting gap from the unstrained material with $C_4$ symmetry through the new smectic phase with two-fold ($C_2$) symmetry and charge-density wave order to a state where superconductivity is completely suppressed.

[1] SUPA, School of Physics and Astronomy, University of St Andrews, North Haugh, St Andrews, Fife KY16 9SS, UK. [2] Department of Physics and Astronomy, University of British Columbia, Vancouver, BC V6T 1Z1, Canada. [3] Stewart Blusson Quantum Matter Institute, University of British Columbia, Vancouver, BC V6T 1Z4, Canada. These authors contributed equally: Chi Ming Yim, Christopher Trainer. Correspondence and requests for materials should be addressed to P.W. (email: wahl@st-andrews.ac.uk)

Since the discovery of striped order in cuprate superconductors[1,2], symmetry breaking, or nematic, electronic states have been found in many strongly correlated electron materials[3–6]. The symmetry breaking states can be classified, in analogy to liquid crystals, into nematic states, which reduce the rotational symmetry without however breaking the translational symmetry, and smectic states which reduce both rotational and translational symmetry[7]. In iron-based superconductors, the nematicity is closely linked to a structural (and often magneto-structural) phase transition into an orthorhombic crystal structure, which exhibits orbital order and an anisotropy of magnetic excitations.[8]

The impact of the lattice anisotropy on the electronic properties of iron-based superconductors in the orthorhombic phase has been studied in great detail, revealing a strong anisotropy of electronic transport[6] and a significant nematic susceptibility even above the orthorhombic phase transition[9–12]. A fundamental question emerging from this is what the influence of small lattice distortions is on the ground state of the materials. Strain-tuning of a material starting from a tetragonal crystal structure has recently been shown for the putative triplet superconductor $Sr_2RuO_4$, revealing a substantial increase in the superconducting transition temperature as a function of uniaxial strain[13,14]. Combining strain tuning with scanning tunnelling microscopy (STM) is highly non-trivial due to the need to prepare atomically clean and flat surfaces in-situ for a sample mounted in a strain device.

The susceptibility of the electronic structure to small amounts of stress applied to the material raises the question of what the impact of small lattice distortions is for superconductivity in these materials. LiFeAs is special among the iron-based superconductors. The material is a superconductor in the undoped, stoichiometric compound and does not exhibit either a structural distortion or magnetic order. LiFeAs has a tetragonal crystal structure, and is therefore ideally suited to study the impact of small lattice distortions on its correlated electronic states.

The superconductivity in iron pnictides is widely believed to be mediated by spin-fluctuation pairing of charge carriers between a hole pocket near the Γ point and electron pockets near the zone corner[15,16]. In this scenario, uniaxial strain (see Fig. 1a) is expected to impact on the near nesting (indicated by arrows in Fig. 1b), rendering the pairing strength anisotropic for strain along [110], with direct consequences for the order parameter[17].

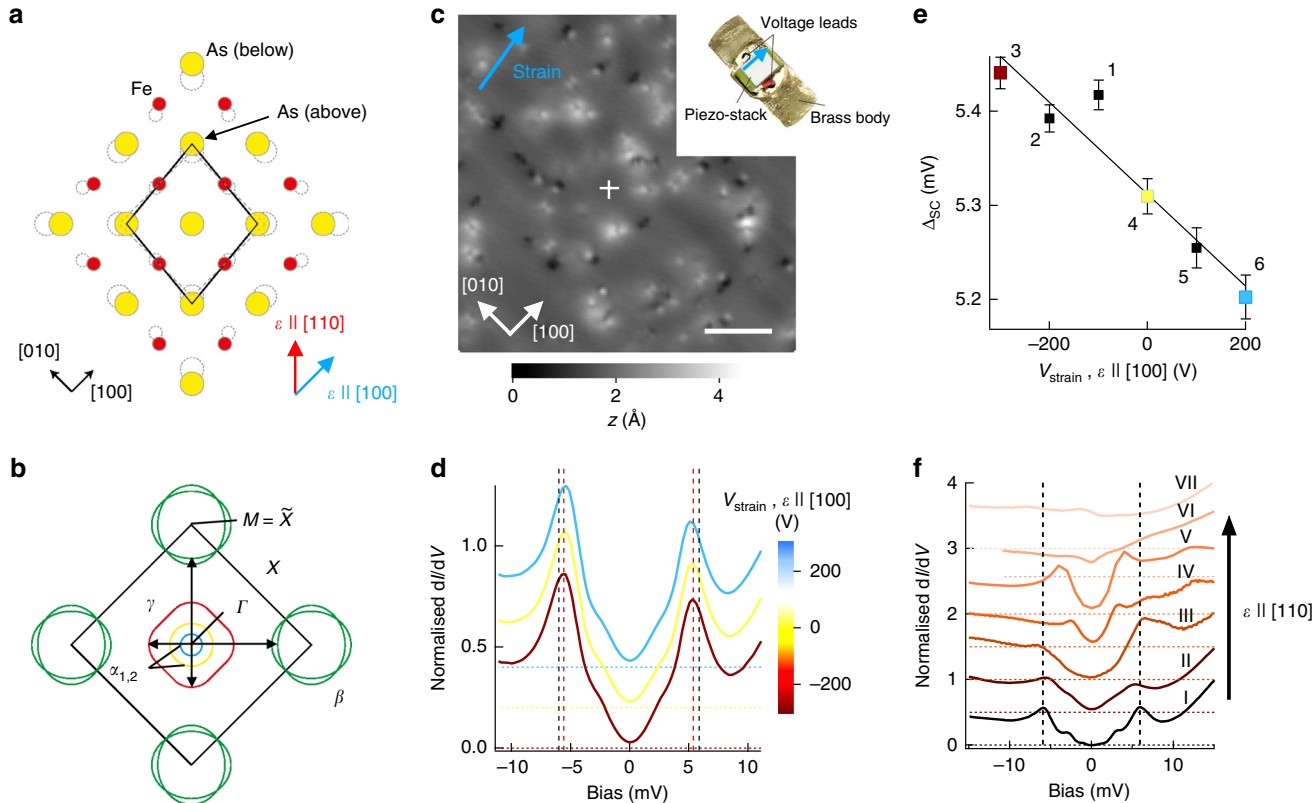

**Fig. 1** Tuning superconductivity in LiFeAs by uniaxial strain. **a** Ball model of the LiFeAs unit cell under positive uniaxial strain along the [110] direction. Dashed open circles represent the atomic positions in the unstrained unit cell, red and blue arrows indicate strain direction along [110] and [100]. **b** Fermi surface of unstrained LiFeAs. Arrows indicate nesting vectors **Q** between hole bands at the zone centre and electron bands at the zone corner. **c** Topographic image of LiFeAs strained along [100] (setpoint conditions $V_s = 20$ mV, $I_s = 50$ pA, $T = 4$ K). A blue arrow indicates the strain direction. Scale bar = 4.3 nm. Inset, sample holder for in-situ strain tuning (without sample). An arrow indicates the direction of strain. **d** d$I$/d$V$ spectra obtained at the position marked with a cross in (**c**) with strain along [100] at different voltages $V_{strain}$ applied to the piezo stack, showing the superconducting gap ($V_s = 14$ mV, $I_s = 0.5$ nA, $T = 4.2$ K, spectra are normalised at $V = 14$ mV). Brown-red dashed vertical lines indicate the positions of the coherence peaks for the spectrum obtained at $V_{strain} = -300$ V, black dashed vertical lines for unstrained LiFeAs. **e** Plot of gap size $\Delta_{SC}$ versus $V_{strain}$ extracted from **d** (see also Supplementary Note 1, Supplementary Figure S7). The number near each data point indicates the order in which the spectra were taken. Error bars of the gap size are obtained from the non-linear least squares fit to the experimental data in **d** and represent the $1\sigma$ confidence interval. For unstrained LiFeAs, $\Delta_{SC} = 5.8$ meV at $T = 4.2$ K, outside the range of the graph. **f** d$I$/d$V$ spectra on samples strained along [110]. Spectra from bottom to top are shown in order of increasing strain. The bottom curve is for an unstrained crystal. All spectra are normalised at $V = 15$ mV and vertically offset for clarity. Vertical lines indicate the energy of the coherence peaks for the unstrained crystal. Horizontal lines in **d**, **f** indicate zero conductance for each of the spectra

For strain along the [100] direction, the consequences are expected to be less dramatic.

Here, we use atomic scale imaging and spectroscopy using low temperature STM with in-situ strain tuning to directly image at the atomic scale the impact of uniaxial strain on the correlated electronic states of the iron-based superconductor LiFeAs. We report on the discovery on a new strain-stabilised smectic state, which exhibits long-range unidirectional charge modulation and coexists with superconductivity.

## Results

**Strain-induced changes in superconductivity in LiFeAs.** To achieve in-situ strain tuning, we have designed a sample holder which enables control of the expansion of a piezo stack to which the sample is mounted, through application of a voltage[18], see inset in Fig. 1c. We have studied the effect of uniaxial strain along both the [100] and [110] directions by mounting samples with different alignments in the device. Upon cooling the sample holder down, the strain in the sample is governed by the aniso-tropic thermal contraction of the piezo stack[19], while the voltage tuning enables a variable additional strain $\delta\varepsilon$ to be applied to the sample. Tunnelling spectra obtained in the same clean spot in the topography shown in Fig. 1c for a sample strained along [100] at different levels of strain are shown in Fig. 1d. All the spectra are characterised by a clear superconducting gap with two pairs of coherence peaks, one near ±5.5 mV and another at ±2 mV. The size of the large gap is significantly smaller than the one observed for unstrained LiFeAs at the same temperature (5.8 mV)[20–23]. Upon application of a voltage to the piezo stack, the field of view moves along the direction of the additional strain (see Supplementary Fig. 1) and a small but systematic change in the size of the superconducting gap, measured at the same defect-free position, is seen (Fig. 1e). As the strain voltage (and $\delta\varepsilon$) along [100] increases, the size of the superconducting gap, extracted from the energies of the outer coherence peaks, is slightly suppressed. The data reveal a direct correlation between the size of the super-conducting gap and the uniaxial strain applied to the sample, demonstrating that we achieve tunability of the superconductivity in LiFeAs.

Strain along the [110] direction has a much larger impact on the spectra and on superconductivity: in Fig. 1f, we show how the tunnelling spectra vary with uniaxial strain along [110], from an unstrained sample (bottom curve) to a sample in which superconductivity is completely suppressed (top curve). We have increased the range of strain achieved by systematically decreas-ing the sample thickness.

**Smectic electronic order in strained LiFeAs.** For intermediate levels of strain (corresponding to Curve V in Fig. 1f), we find that the material enters into a smectic phase where not only the rotational symmetry is reduced from $C_4$ to $C_2$, but also the translational symmetry is broken through a long-range spatial modulation of the charge density. Figure 2a shows a topographic STM image of this phase revealing stripe-like patterns. The appearance of the charge modulation exhibits a phase shift between positive and negative bias voltages (Fig. 2a, b), a key signature for a charge-density wave (CDW). For comparison, STM images of unstrained LiFeAs are shown in Fig. 2c, exhibiting no trace of this modulation, consistent with previous studies[20–23]. The stripes in the modulated phase run along the [110] crystal-lographic direction of LiFeAs, parallel to the direction of the applied strain. They have a spatial periodicity of 2.7nm (Supplementary Note 2, Supplementary Fig. 2), corresponding to a wave vector of $q \approx 0.14q_0$, independent of the applied strain. A closer inspection reveals that the stripes exhibit topological

defects (Supplementary Fig. 3). Topological defects play an important role in coupling nematic and smectic orders, and their observation suggests that the uniaxial strain drives the material through a nematic phase into the smectic order.

To assess whether the modulation of the charge density is associated with a characteristic energy scale, and its influence on superconductivity, we have acquired a spectroscopic map to study the electronic states across the modulation. Figure 2d, e shows the topography and a differential conductance map $g(\mathbf{x}, V) = dI/dV$ ($\mathbf{x}, V$). The map exhibits a strong modulation of the height of the superconducting coherence peaks, which can be more clearly seen from spectra taken on top of the charge modulation and between the maxima in Fig. 2f. Most notably, the spectra show an additional feature at 12 mV which is modulated with opposite phase compared to the coherence peaks. In addition, a weak feature can be seen at −18 mV. To extract the characteristic energy scale of the charge modulation, we analyse the amplitude of the modulation in the ratio of the differential to the total conductivity $l(\mathbf{x}, V) = g(\mathbf{x}, V)/(I(\mathbf{x}, V)/V)$, a quantity for which the set point effect due to variation in the tip-sample distance is suppressed if the tunneling matrix element is only weakly energy dependent and which can be taken as a representative of the density of states $\rho(\mathbf{x}, V)$[24,25]. From the above, we calculate at each bias voltage the spatial variance of $l(\mathbf{x}, V)$, denoted $\sigma^2(l(\mathbf{x}, V))$, to determine what the characteristic energy scale of the charge modulation is. In Fig. 2g we show the variance in $l(\mathbf{x}, V)$ of the stripe modulation, as a function of bias voltage, as well as its wave vector. The variance exhibits a clear maximum at +11 mV and −16 mV, at slightly smaller energies than the maxima in $g(V)$ seen in Fig. 2f. The wave vector stays practically constant within the energy range investigated here, confirming that it stems from a static charge modulation rather than quasi-particle interference, which would lead to a dispersion of the modulation. From the contrast inversion seen in topographic images and the character-istic energy scale of the modulation we observe in the differential conductance $g(\mathbf{x}, V)$ (or $l(\mathbf{x}, V)$), we attribute the modulated phase to the formation of a CDW, with formation of a partial gap between +11mV and -16mV.

In this CDW phase, the shape of the superconducting gap differs significantly from the one found in unstrained LiFeAs, as can be seen from the spectra in Fig. 2f. Spectra taken in the CDW phase are characterised by a pair of superconducting coherence peaks at ±4 mV. The gap size is substantially reduced compared to unstrained LiFeAs[21].

**Vortex lattice and vortex core bound states in the smectic phase.** In order to assess how the modulation of the charge density affects the superconducting state we have studied the vortex lattice and vortex core bound states in this phase. Fig-ure 3a, b shows the topography and map of the zero-bias con-ductance of strained LiFeAs in a magnetic field. The topographic appearance of the modulation (Fig. 3a) is unaffected by the magnetic field, whereas the vortex cores appear distorted by the stripe modulation, and their whole appearance becomes modu-lated and elongated in the direction perpendicular to the stripes. For comparison, we show the vortex lattice of unstrained LiFeAs in Fig. 3c. Most intriguingly, the charge modulation changes the vortex core bound state: while in unstrained LiFeAs, the vortex cores exhibit a bound state peak centred at −0.6 mV[22], in the modulated phase the cores exhibit a bound state peak at +0.6 mV (Fig. 3d). The particle-hole asymmetry of the vortex core bound states has been ascribed previously to vortices being close to the quantum limit, where the coherence length $\xi$ is of the order of the Fermi length, $k_F\xi \sim 1$[22,26–28]. The change of the dominant bound state energy from the occupied states in unstrained LiFeAs to the

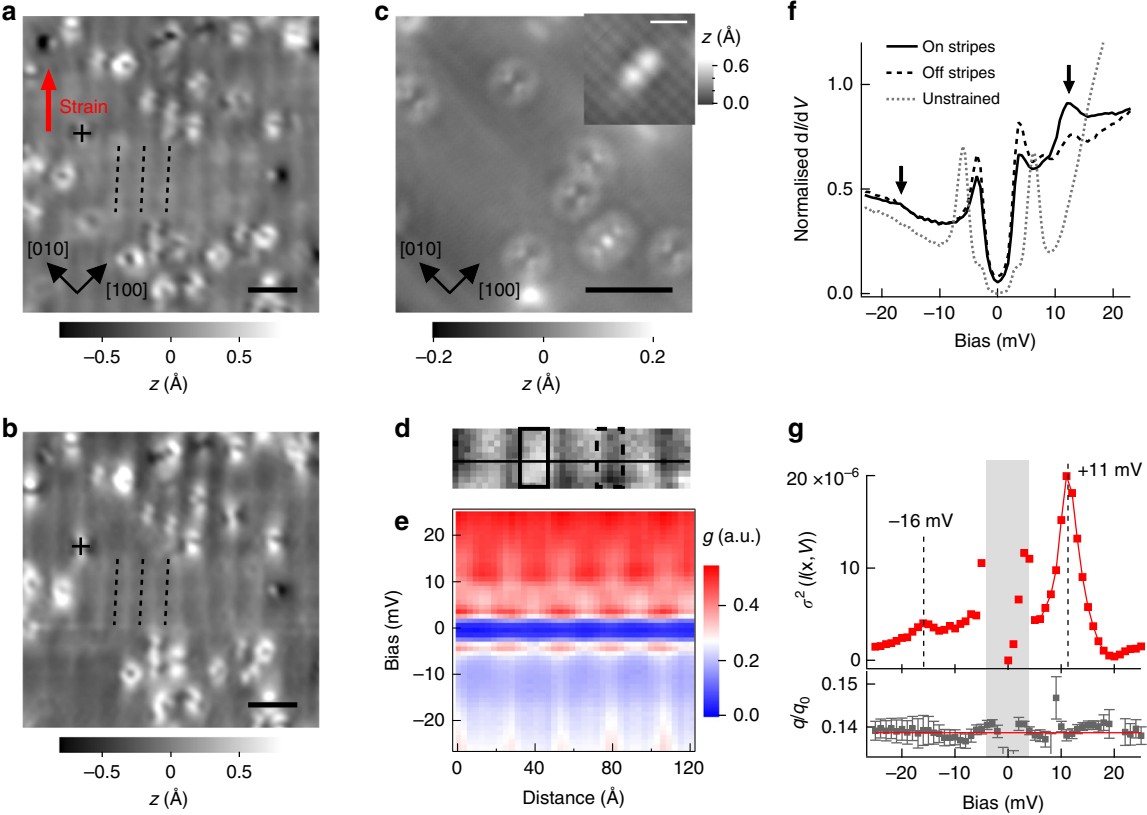

**Fig. 2** Modulated phase of LiFeAs strained along [110]. **a, b** Topographic images taken from the same position on the modulated phase at **a** $V_s = +17$ mV and **b** $V_s = -18$ mV (both $I_s = 50$ pA). A red arrow indicates the direction of the strain. Crosses mark the position of the same point defect. Dashed lines highlight the phase shift of the modulation between the two images. **c** Topographic image ($V_s = +15$ mV, $I_s = 0.25$ nA) and (inset, $V_s = -50$ mV, $I_s = 0.3$ nA) an atomically resolved image of unstrained LiFeAs. Scale bars in **a–c** 5 nm, in inset of **c** 1 nm. **d** Topography of the modulated phase (12.4 × 3.1 nm², $V_s = 25$ mV, $I_s = 0.1$ nA). **e** Differential conductance $g(\mathbf{x}, V)$ as a function of position and bias voltage across the stripes, along the solid line in **d** ($V_s = 25$ mV, $I_s = 0.3$ nA). **f** d$I$/d$V$ spectra recorded from the central positions on and off the stripes (areas indicated by solid and dashed rectangle in **d**, respectively; $V_s = 30$ mV, $I_s = 0.5$ nA). Arrows mark the bias voltages where the contrast of the charge modulation in **e** is strongest. A spectrum obtained on unstrained LiFeAs (dashed grey line, $T = 1.5$ K, $V_s = -50$ mV, $I_s = 0.3$ nA) is included for comparison. The spectra were normalised at $V = 30$ mV. Unless stated otherwise, all data in **a,b, d–f** have been recorded at $T = 4$ K. **g** (Top panel) Spatial variance $\sigma^2(V) = \left\langle I(\mathbf{x}, V)^2 \right\rangle_{\mathbf{x}} - \left\langle I(\mathbf{x}, V) \right\rangle_{\mathbf{x}}^2$ of the calculated normalised conductance $I(\mathbf{x}, V) = g(\mathbf{x}, V)/(I(\mathbf{x}, V)/V)$ as a function of bias voltage, extracted from **e**. Dashed vertical lines mark the positions of the charge modulation peaks. (Bottom panel) Wave-vector of the spatial modulation of $I(\mathbf{x}, V)$ as a function of bias voltage. Error bars are obtained from the non-linear least squares fit to the experimental data in **e** and represent the $1\sigma$ confidence interval. A red line shows the average wave vector of $0.139q_0$. Inside the superconducting gap (shaded grey), $I(\mathbf{x}, V)$ becomes unreliable because the current becomes very small

unoccupied states in the strained sample indicates that either the superconducting gap on the small hole pocket at the Γ-point (labelled α in Fig. 1b) is strongly suppressed, or that the pocket is reconstructed by the modulated phase. This is also consistent with the reduced size of the superconducting gap, as the largest gap has been reported for the α-band at the Γ point for unstrained LiFeAs[29].

Temperature dependent measurements show that the superconducting transition temperature is suppressed to about 13K in the modulated phase (see Fig. 4a), consistent with the reduced size of the superconducting gap. The modulated phase persists into the normal state (Supplementary Fig. 4), demonstrating that it emerges from the normal state and superconductivity forms on top of it.

## Discussion

The picture which emerges from our measurements is summarised in the phase diagram in Fig. 4b. Starting from the unstrained material, superconductivity is initially suppressed slightly with increasing strain. At intermediate strain, the material enters into the CDW phase. Images of coexistence between areas

showing the modulated phase and areas with no modulation suggest that this is a first order phase transition. Phase coexistence has also been found in biaxially strained iron pnictides[30]. At the transition to the modulated phase, the size of the superconducting gap is reduced rather abruptly, with spectra of areas of the sample which are in the modulated phase showing a significantly smaller gap than areas which have not undergone the transition. The modulated phase itself is hardly influenced by additional strain, and retains the same periodicity. The intensity with which we observe the modulation is reduced with increasing strain until it is completely suppressed (Supplementary Fig. 5). Superconductivity is suppressed (at the temperature of our measurements) at the same level of strain where the modulation of the charge density disappears. At higher levels of strain we see no trace of the stripe-like modulation or of superconductivity.

Several mechanisms could lead to the formation of a unidirectional modulation of the density of states. The magnetic order observed in several iron pnictides[31], as well as the spin fluctuations, are unlikely candidates as they occur at a completely different wave vector at (or near) $\mathbf{q} = (1/2, 1/2)$. A peculiarity in the spin excitations in LiFeAs is that the wave vector of the spin resonance mode is incommensurate, with an incommensurability

of $\delta = 0.07$[32], resulting in a splitting in the maxima by about the same wave vector as the modulation reported here. This indicates an instability in the susceptibility towards the modulated order we see here.

Nematic orders in iron pnictides normally break the rotational symmetry but not translational symmetry: they do not by themselves lead to an additional periodicity[5,33–35], but rather a strong anisotropy of the electronic structure. However, nesting in a nematic phase might lead to a modulation of the charge density and hence a smectic electronic state as observed here. Formation of such a CDW could be additionally stabilised by a lattice instability due to a softening of a phonon mode. Calculations suggest that the phonon dispersion does exhibit a minimum at the zone face[36].

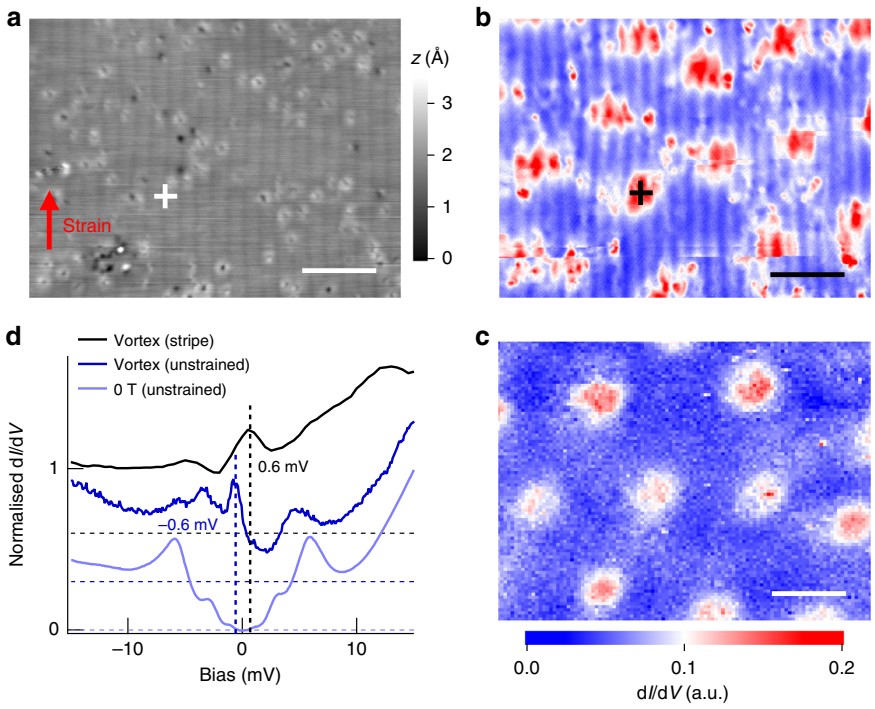

**Fig. 3** Vortices in the modulated phase. **a** Topographic image of the modulated phase taken in the presence of 9 T out-of-plane field at $T = 4$ K ($V_s = 25$ mV, $I_s = 50$ pA). A red arrow indicates the direction of the strain. **b** Corresponding $dI/dV$ map recorded at zero bias. Vortices are seen as areas of high zero bias conductance (red). **c** $dI/dV$ map taken at bias voltage of $-0.5$ mV on an unstrained sample in the presence of an out-of-plane field of 5T at 1.2 K. Scale bars in **a–c**: 12.5 nm. **d** $dI/dV$ spectrum taken in the centre of one of the vortices formed on the modulated phase (black, the position is marked with a cross in **a**, **b** and those on the surface of an unstrained LiFeAs crystal (mid-blue), respectively. An averaged spectrum taken from a defect-free region on unstrained LiFeAs at 0T, 1.2 K (light-blue) is included for reference. Spectra are vertically offset for clarity. Dashed horizontal lines indicate the positions of zero conductance for each spectrum. For **b**, **c**, $V_s = 25$ mV, $I_s = 0.3$ nA

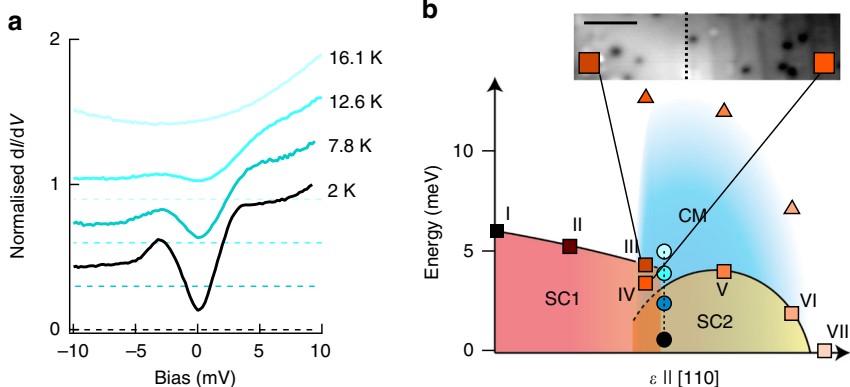

**Fig. 4** Phase diagram of LiFeAs as a function of uniaxial strain. **a** $dI/dV$ spectra acquired on LiFeAs in the modulated phase at different temperatures (all spectra normalised at $V = 10$ mV, Dashed horizontal lines are zero conductance for each spectrum, $V_s = 10$ mV, $I_s = 0.15$ nA). **b** Phase diagram of LiFeAs as a function of uniaxial strain along [110], with the superconducting gap size (squares) and the characteristic energy of the CDW (triangles) as a function of strain. STM images showing coexistence between SC1 and CM/SC2 (see inset) indicate a first order phase transition to the modulated phase. Roman numerals and colours of squares and triangles refer to spectra in Fig. 1f. Round points corresponding to spectra in **a** are positioned at $3.57 k_B T$, using $\Delta / k_B T_c$ determined from the temperature dependence in **a**. Inset: topographic image of an area showing coexistence of SC1 and SC2/CM phases. Scale bar: 6 nm. A large-scale topographic image showing the coexistence of the two phases is shown in Supplementary Fig. 6

Our results report the discovery of a strain-induced transition into a CDW phase for a material which in its ground states does not show any evidence for nematicity, charge order or magnetic order. They provide surprising new evidence for the importance of the coupling between the strongly correlated electronic states and lattice degrees of freedom. We introduce strain STM as a new tool to stabilise and visualise novel superconducting phases and the interplay between electronic correlation effects and lattice distortions at the atomic scale.

## Methods

**STM measurements**. The STM experiments were performed using two home-built low temperature STMs which operate at a base temperature of 1.8 K[18,37]. Pt-Ir tips were used, and conditioned by field emission with a Au sample. Differential conductance (d$I$/d$V$) maps and single point spectra were obtained using a standard lock-in technique, with frequency of the bias modulation set at 409.2 Hz. To obtain fresh and clean surfaces for STM measurements, 0.5% Zn- (Co-) doped LiFeAs samples were cleaved in-situ at ~20 K in cryogenic vacuum. The results reported here have been obtained from a total of 20 cleaves for unstrained LiFeAs and 11 cleaves of four different samples for strained LiFeAs with sample thickness ranging from 0.1 to 0.4mm.

**Piezoelectric device**. Motivated by previous nematic susceptibility measurements by Chu et al.[10], we have constructed a sample holder which allows for the application of in-situ tunable strain to single crystal samples. As shown in Fig. 1c in the main text, the sample holder comprises a brass main body, and a piezoelectric actuator which is mounted side-ways atop the main body. By applying a positive (negative) voltage across the leads of the piezoelectric actuator, it expands (contracts) along the longitudinal direction and contracts (expands) along the transverse direction. Samples were glued onto the side-wall of the piezoelectric stack facing towards the STM tip using Epotek H20E conductive epoxy. The orientation of the crystal relative to the piezoelectric stack determines the direction in which the strain is applied. Samples were cleaved by glueing a rod on top of the sample, which was knocked off at an in-situ cleaving stage[37]. To extend the range of strain achieved at the surface of the material, we have studied multiple cleaves of the same sample, as the strain detected at the surface depends on the sample thickness. For the sample strained along [100], the main direction of the strain was within 20° of the [100] direction, for the [110] the alignment was better than 5°. Anisotropic thermal contraction/expansion of the piezo stack leads to a strain at the interface between stack and the LiFeAs sample of about 0.3%, which provides an upper boundary for the levels of strain achieved here. Strain levels achieved by voltage tuning were up to 0.01% at the interface.

**Sample growth**. LiFeAs samples were grown using a self-flux technique[21]. Samples studied here contained minute amounts of engineered defects such as Zn and Co (on the order of 0.5%), which do not affect the spectra of clean areas of the surface or the occurrence of the modulated phase[38].

**Data availability**. Data are available online.[39]

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

## Acknowledgements

C.T., C.M.Y., and P.W. acknowledge funding from EPSRC through EP/L505079/1 and
EP/I031014/1. Research at UBC was supported by the Natural Sciences and Engineering
Research Council of Canada, the Canadian Institute for Advanced Research, and the
Stewart Blusson Quantum Matter Institute. We acknowledge valuable discussions with
Clifford Hicks, Andreas Kreisel, Andreas Rost, Steve Simon, and Matt Watson.

## Author contributions

C.M.Y. and C.T. performed experiments on strained LiFeAs samples and analysed the
data. R.A. and S.C. performed STM measurements on unstrained LiFeAs. S.C., W.N.H.,
R.L., and D.A.B. grew the crystals. C.M.Y. and P.W. wrote the manuscript. All authors
discussed and contributed to the manuscript.

## Additional information

018-04909-y.

**Competing interests:** The authors declare no competing interests.

**Reprints and permission** information is available online at http://npg.nature.com/
reprintsandpermissions/

**Publisher's note:** Springer Nature remains neutral with regard to jurisdictional claims in
published maps and institutional affiliations.

