## [Peer Review File · Nature Communications]

Reviewers' comments:

Reviewer #1 (Remarks to the Author):

The authors measured the topography and the tunneling spectra on LiFeAs under an unidirectional strain. The iron based superconductor LiFeAs is supposed to be tetragonal with C4 symmetry of electronic structure. Now under an in-plane unidirectional strain, it seems that a nematic state has been induced in LiFeAs and detected by STM/STS measurements. They find that the superconductivity gap is suppressed in the strained state, with the strongest suppression with the strain along [010] direction. They thus observed a CDW phase in the strained state. They also determined the CDW energy at about -14 and 16 meV. Although the nematic state has been reported in many iron based superconductors, including studies of STM/STS, I still believe this paper contains some new message which are useful for the community. The proposed method of strain STM measurements are also quite intriguing and useful, therefore I would recommend a possible publication of this work. However, before making a final recommendation, I would like invite the authors taking the following concerns into account.

1. What plotted in Fig.4 suggest that a first order transition occurs at certain strain. Since STM/STS are local measurements, I just wonder whether the C2 modulation starts already at a much lower strain in some small areas and they did not touch those areas. In addition, it would be better to show one figure within one FOV, in some area the C2 phase shows up, but in some other areas the C2 phase is still absent. Otherwise the phrase "mixture of the two phases" seem finding no support.

2. From the physical point of view, CDW and nematic phase are different terminologies. It requires more elaboration why the authors regard this C2 as CDW phase, not the widely used name "nematicity" in this strained phase.

3. Concerning the vortex core bound state, it seems the vortex core has been sliced to many parallel segments. I wonder at which point the spectrum in Fig.3d was measured, at the ridge of the stripe, or valey? Or it detects a averaged signal. I believe the spectra at the ridge and the valey should be very different.

Reviewer #2 (Remarks to the Author):

The authors here present a detailed STM study on a strained Fe-based superconductor LiFeAs. They show the superconductivity can be tuned by the strain. Furthermore, a stable CDW phase appears with increasing strain while the SC is suppressed, which suggests the possible ground state for strained LiFeAs. The wavevector of the CDW modulation seems to be related to the incommensurability observed in neutron experiments. Their results show the importance of lattice degree freedom in high T_c superconductivity. Using the piezo to strain or detwin the single crystals is common but remains challenging for spectroscopy measurements, especially STM. Their work will surely promote further study by using strain as new tuning parameter for high T_c superconductivity. Thus, I will recommend this work to be published in Nature Communication but with some revisions and suggestions.

(1) The data presented here are of great interest and with high quality. However, I feel the introduction (the first paragraph in P.2) is too simplified for me. I think the readers can benefit from a more detailed introduction, especially for those who are not following closely with the anisotropic electronic states in the orthorhombic phase of Fe-based superconductor. It may also be helpful to mention the detailed difference between LiFeAs and other Fe-based superconductors.

(2) For the results in Fig.2, the authors states that “the material enters into a nematic phase which exhibits a long-range spatial modulation of the charge density” (nematic also appears later in the discussion). Since there’s broken translational symmetry with a periodicity $\sim 2.7\text{nm}$ in their data and the strain doesn’t change the periodicity, this should be smetic.

(3) The observation of topological defects in these stripes is very interesting as they can greatly affect the liquid crystal structures and dynamics (likely so for the electronic counterpart). Although it’s not the main point of their story, the lack of any description at the end of this paragraph is very odd.

(4) The Feenstra normalization $I(x,V)$ is not completely free from the setup effect as the authors claim. The variance should also be defined clearly in Fig2g.

(5) It will be interesting to see if the bosonic mode coupling can be tuned by the strain. I am wondering if the analysis such as PRL109.087002 can be done on the strained spectrum.

(6) The vortex cores are strongly modulated by the charge stripes in Fig3b. Where is the spectrum in Fig3d taken or is it the averaged one?

Reply to Referees

We thank all referees for their time and careful review of our manuscript, as well as for their constructive criticism. We address their queries below and are confident that we have addressed all pertinent issues. Our responses are typeset in *italic*.

Reviewer #1 (Remarks to the Author):

The authors measured the topography and the tunneling spectra on LiFeAs under an unidirectional strain. The iron based superconductor LiFeAs is supposed to be tetragonal with C4 symmetry of electronic structure. Now under an in-plane unidirectional strain, it seems that a nematic state has been induced in LiFeAs and detected by STM/STS measurements. They find that the superconductivity gap is suppressed in the strained state, with the strongest suppression with the strain along [010] direction. They thus observed a CDW phase in the strained state. They also determined the CDW energy at about -14 and 16 meV. Although the nematic state has been reported in many iron based superconductors, including studies of STM/STS, I still believe this paper contains some new message which are useful for the community. The proposed method of strain STM measurements are also quite intriguing and useful, therefore I would recommend a possible publication of this work. However, before making a final recommendation, I would like invite the authors taking the following concerns into account.

1. What plotted in Fig.4 suggest that a first order transition occurs at certain strain. Since STM/STS are local measurements, I just wonder whether the C2 modulation starts already at a much lower strain in some small areas and they did not touch those areas. In addition, it would be better to show one figure within one FOV, in some area the C2 phase shows up, but in some other areas the C2 phase is still absent. Otherwise the phrase "mixture of the two phases" seem finding no support.

We thank the referee for his useful comment. STM is a local probe and only probes a fractional area of the sample. It is therefore possible that the modulated phase starts to form locally at lower strain without being seen in our STM images. To reflect this possibility, as well as the first order nature of the phase transition, we have modified the phase diagram in Fig. 4, so that there is now some overlap between the SC1 and SC2/CM phases at intermediate strain to indicate a coexistence region.

In the original manuscript we had included an STM image (inset of Fig. 4b) which is very similar to that suggested by the referee. We agree that a clearer illustration on their coexistence will enhance the readability of the manuscript. Therefore, we have included

a larger scale STM image showing the coexistence of the two phases as Supplementary Fig. 6.

2. From the physical point of view, CDW and nematic phase are different terminologies. It requires more elaboration why the authors regard this C2 as CDW phase, not the widely used name "nematicity" in this strained phase.

We agree with the referee, and following his and the second referees suggestion we now refer to the CDW phase with C2 symmetry as a smectic phase, consistent with the terminology introduced in ref. 7 of the revised manuscript.

3. Concerning the vortex core bound state, it seems the vortex core has been sliced to many parallel segments. I wonder at which point the spectrum in Fig.3d was measured, at the ridge of the stripe, or valley? Or it detects an averaged signal. I believe the spectra at the ridge and the valley should be very different.

The spectrum is indeed an averaged point spectrum recorded right at the center of one of the vortex cores on the modulated phase. The position is now marked with a cross in the FOV in the revised Fig. 3, a and b. We do not have systematic spectroscopic data obtained at positions at the ridge/valley across the vortex core, however from analyzing the spatial map there is no evidence for a strong change in the spectra.

Reviewer #2 (Remarks to the Author):

The authors here present a detailed STM study on a strained Fe-based superconductor LiFeAs. They show the superconductivity can be tuned by the strain. Furthermore, a stable CDW phase appears with increasing strain while the SC is suppressed, which suggests the possible ground state for strained LiFeAs. The wavevector of the CDW modulation seems to be related to the incommensurability observed in neutron experiments. Their results show the importance of lattice degree of freedom in high T_c superconductivity. Using the piezo to strain or detwin the single crystals is common but remains challenging for spectroscopy measurements, especially STM. Their work will surely promote further study by using strain as a new tuning parameter for high T_c superconductivity. Thus, I will recommend this work to be published in Nature Communication but with some revisions and suggestions.

(1) The data presented here are of great interest and with high quality. However, I feel the introduction (the first paragraph in P.2) is too simplified for me. I think the readers can benefit from a more detailed introduction, especially for those who are not following closely with the anisotropic electronic states in the orthorhombic phase of Fe-based superconductor. It may also

be helpful to mention the detailed difference between LiFeAs and other Fe-based superconductors.

We have revised the introduction, with additional references and introducing nematicity in the broader context of strongly correlated electron materials. We have also introduced a paragraph which highlights the key differences between LiFeAs and the other Fe-based superconductors.

(2) For the results in Fig.2, the authors states that “the material enters into a nematic phase which exhibits a long-range spatial modulation of the charge density” (nematic also appears later in the discussion). Since there’s broken translational symmetry with a periodicity $\sim 2.7\text{nm}$ in their data and the strain doesn’t change the periodicity, this should be smectic.

We appreciate the referee’s comment on this, and in fact while preparing the manuscript we had been considering the same question. We have followed the suggestion of the referee and changed the wording to smectic where we refer to the CDW phase we report here.

(3) The observation of topological defects in these stripes is very interesting as they can greatly affect the liquid crystal structures and dynamics (likely so for the electronic counterpart). Although it’s not the main point of their story, the lack of any description at the end of this paragraph is very odd.

We are grateful for the referee to point out this point, and have included a few more sentences to discuss the relevance of topological defects in coupling smectic and nematic orders.

(4) The Feenstra normalization $I(x,V)$ is not completely free from the setup effect as the authors claim. The variance should also be defined clearly in Fig2g.

The Feenstra’s normalisation may not completely remove the setpoint effect in the spectroscopic data if the tunneling matrix element is voltage dependent. However, in a small bias voltage range and assuming that the tunneling matrix element does not vary strongly, the influence of the setup condition is strongly suppressed. The main disadvantage is that $I(x,V)$ is not proportional to the local density of states $\rho(x,V)$ anymore. Normalising our data this way helps to identify the energies of strongest contrast in the modulated phase from the spectroscopic-imaging data without artifacts due to the setpoint condition and to identify the contrast reversal of the modulated phase

as the sample bias crosses the energy position of the CDW peak. This is confirmed by the contrast reversal seen in topographic images, confirming the analysis (see fig. 2a and b). We have added a sentence which defines the variance in the revised manuscript, as well as the equation to calculate σ^2 in the caption.

(5) It will be interesting to see if the bosonic mode coupling can be tuned by the strain. I am wondering if the analysis such as PRL109.087002 can be done on the strained spectrum.

We appreciate the referee's suggestion. The extraction of the bosonic mode information from the tunneling spectra is a non-trivial task, requiring all the spectra to be normalised by the normal state spectrum, usually obtained at 20 K. We have not obtained spectroscopy data at 20 K for all the strain values. Moreover, it happens that the bosonic mode in the strained sample (at ~12 mV above the Fermi level) is expected very closely in energy to the CDW peaks of the modulated phase, which are significantly more pronounced in the tunneling spectra. Disentangling the two contributions from one another is therefore not without ambiguity.

(6) The vortex cores are strongly modulated by the charge stripes in Fig3b. Where is the spectrum in Fig3d taken or is it the averaged one?

The spectrum is indeed an averaged point spectrum recorded right at the center of one of the vortex cores on the modulated phase. The position is now marked with crosses in the FOV in the revised Fig. 3, a and b.

Summary of changes:

Main text

- *Shortened abstract to conform to Nature Communications style guide*
- *Revised the introduction to introduce nematicity in the broader context of strongly correlated electron materials, and to briefly highlight the key differences between LiFeAs and the other Fe-based superconductors (see highlighted text in page 2), adding references 1-8, as well as more discussion of strain tuning of correlated electron materials.*
- *Added a sentence defining the variance as suggested by referee 2 (see highlighted text in page 4), as well as the equation in the caption of fig. 2*
- *Added a sentence on the importance of topological defects (see highlighted text on pg. 4)*

Figures:

- *Changed the color code in Fig. 1, d and e to avoid red and green colors being used in the same plot.*
- *Replaced the original Fig. 3 with the revised version, to indicate the position in the FOV at which the dI/dV spectrum in the vortex core was taken, with corresponding change in the figure caption*
- *Replaced the original Fig 4 with a revised version, in which the SC1 and SC2 phases in the phase diagram in Fig. 4b overlap with each other at intermediate strain*

Supplementary Information

- *Added new Supplementary Fig. 6 showing a STM image of the coexistence between the SC1 and SC2 phases at intermediate strain in the same field-of-view to the revised Supplementary Information*
- *Removed Supplementary Figures which are neither referenced in the main text nor in the Supplementary Information.*
- *Corrected typo in eq. 1.*

REVIEWERS' COMMENTS:

Reviewer #1 (Remarks to the Author):

I think the authors have given sufficient addressing to my concerns and made related revisions. I have no more concerns and thus recommend the acceptance to the paper.

Reviewer #2 (Remarks to the Author):

Authors have made necessary changes that take into account all suggestions and answered to all comments. I believe their results are interesting and relevant. This is an important step toward the microscopic understanding of the interplay between the lattice degree of freedom and the high T_c superconductors. Thus, I recommend its publication in Nature Communication.